# Protein Tyrosine Phosphatase Receptor Type Z in Central Nervous System Disease

**DOI:** 10.3390/ijms23084414

**Published:** 2022-04-16

**Authors:** Kenichiro Nagai, Masazumi Fujii, Shinobu Kitazume

**Affiliations:** 1Department of Neurosurgery, School of Medicine, Fukushima Medical University, Fukushima 960-1295, Japan; longwell@fmu.ac.jp (K.N.); fujiim@fmu.ac.jp (M.F.); 2Department of Clinical Laboratory Sciences, School of Health Sciences, Fukushima Medical University, Fukushima 960-8516, Japan

**Keywords:** PTPRZ, phosphacan, glioma, demyelination, multiple sclerosis, diagnostic marker, drug discovery, chondroitin sulfate

## Abstract

Gliomas are among the most common tumors of the central nervous system and include highly malignant subtypes, such as glioblastoma, which are associated with poor prognosis. Effective treatments are therefore urgently needed. Despite the recent advances in neuroimaging technologies, differentiating gliomas from other brain diseases such as multiple sclerosis remains challenging in some patients, and often requires invasive brain biopsy. Protein tyrosine phosphatase receptor type Z (PTPRZ) is a heavily glycosylated membrane protein that is highly expressed in the central nervous system. Several reports analyzing mouse tumor models suggest that PTPRZ may have potential as a therapeutic target for gliomas. A soluble cleaved form of PTPRZ (sPTPRZ) in the cerebrospinal fluid is markedly upregulated in glioma patients, making it another promising diagnostic biomarker. Intriguingly, PTPRZ is also involved in the process of remyelination in multiple sclerosis. Indeed, lowered PTPRZ glycosylation by deletion of the glycosyltransferase gene leads to reduced astrogliosis and enhanced remyelination in mouse models of demyelination. Here, we review the expression, molecular structure, and biological roles of PTPRZ. We also discuss glioma and demyelinating diseases, as well as the pathological role of PTPRZ and its application as a diagnostic marker and therapeutic target.

## 1. Introduction

Protein tyrosine phosphatase receptor type Z (PTPRZ), also called RPTPβ, is a membrane-bound protein with tyrosine phosphatase activity localized to its cytoplasmic region. PTPRZ is highly expressed in glial cells of the central nervous system (CNS), including oligodendrocyte precursor cells (OPCs), astrocytes, and oligodendrocytes [1]. PTPPRZ is associated with the regulation of the differentiation of OPCs into oligodendrocytes for myelination in normal brain tissue [2,3], and there is accumulating evidence for its involvement in CNS diseases. Several reports have shown that PTPRZ is abundantly expressed in gliomas [4,5]. Gliomas are the most common and most challenging primary brain tumor; accordingly, much research has been devoted to its treatment. PTPRZ is also involved in remyelination in multiple sclerosis (MS), the most common demyelinating disease of the CNS [6]. PTPRZ in the CNS undergoes brain-specific glycosylation with moieties such as chondroitin sulfate (CS), keratan sulfate, and branched O-mannosyl (Man) glycans. Compared with normal brain tissues, gliomas express heavily glycosylated PTPRZ. The glycosyl modifications play important roles in the demyelination processes in MS and tumor cell growth in gliomas. In this review, we describe the recent findings on both the physiological and pathological roles of PTPRZ, and we discuss its potential applications in the treatment of MS and gliomas.

## 2. Molecular Structure and Physiological Roles of PTPRZ

PTPRZ is a member of the receptor-like protein tyrosine phosphatase (RPTP) family of proteins, which comprises structurally and functionally diverse enzymes grouped into eight subfamilies [7]. Both PTPRZ and PTPRG have an extracellular carbonic anhydrase (CAH)-like domain, a fibronectin-like domain, and two intracellular tyrosine phosphatase domains, and are categorized into the R5 subtype based on sequence similarity [8]. PTPRG exhibits relatively broad tissue distribution, while PTPRZ is mainly expressed in the CNS [1], with highest expression in astrocytes, intermediate expression in oligodendrocyte precursor cells (OPCs) and oligodendrocytes, and limited expression in neurons [9,10]. PTPRZ has several isoforms derived from alternative mRNA splicing. In humans, these isoforms are divided into two groups, PTPRZ-long and PTPRZ-short, both of which have a transmembrane domain, but with the latter lacking a juxtamembrane extracellular region (~900 amino acids) (Figure 1). In mice, an mRNA isoform corresponding to soluble PTPRZ lacking a transmembrane region, named phosphacan, is also present [6]. PTPRZ expressed in the CNS is highly modified with chondroitin sulfate (CS) [6,11,12], keratan sulfate [13], and branched O-Man glycans [14,15] capped with HNK-1 [16] or Lewis X epitopes [17]. In comparison, PTPRZ detected in peripheral tissues, such as gastric glands, lacks proteoglycans [18,19].

PTPRZ does not appear to play an essential role in mammalian development, as mice lacking it develop normally [20]. Several studies have been carried out on the regulatory function of PTPRZ in the differentiation of OPCs into oligodendrocytes and its roles in myelination and learning and memory [6,21]; however, the underlying molecular mechanisms remain unclear. PTPRZ functions as a cell adhesion molecule, and binds with the neural recognition molecule contactin through its CAH-like domain to control the development of OPCs [2,3]. PTPRZ also functions as a signaling molecule, and several groups have reported that the binding of the cytokine pleiotrophin (PTN) to the extracellular PTPRZ region promotes head-to-toe dimer formation, causing steric hindrance of the catalytically active PTP D1 domain by the PTP D2 domain, and leading to the inactivation of phosphatase activity (Figure 2) [22,23,24]. Other known PTPRZ inhibitory ligands include midkine (MK) and IL-34 [25,26]. The CS chains attached to PTPRZ are suggested to maintain PTPRZ in a monomeric active state until ligand binding [27], while the branched O-Man glycan promotes the dimerization of PTPRZ, thereby inhibiting phosphatase activity [14]. The proto-oncogene Fyn is highly overexpressed in the brain and phosphorylates downstream signaling molecules, such as β-catenin and p190 RhoGAP [28,29], which are dephosphorylated by PTPRZ [27,30]. Nevertheless, it seems unlikely that protein tyrosine kinase (PTK) and PTP modulate cell signaling in a simply reversible manner, as inhibiting either Fyn or PTPRZ results in cell growth and migration.

## 3. PTPRZ and Glioma

### 3.1. What Are Gliomas?

Gliomas are one of the most common brain tumors, accounting for approximately 30% of all primary CNS tumors [31,32]. The term “glioma” originated from the histomorphological similarities between tumor cells and the glial cell lineage in the normal brain, including astrocytic and oligodendroglial cells, and designates a category of various glial cell tumors. The majority are “diffuse gliomas”, which are malignant and infiltrative, and impair brain functions, often causing seizures, and eventually lead to death. Among diffuse gliomas, glioblastomas are one of the most malignant cancers in humans, with a life expectancy of 15 months regardless of treatment with surgery, chemotherapy, or radiotherapy [33]. There are still many challenges in the diagnosis and treatment of gliomas. Despite recent advances in diagnostic imaging modalities, such as CT, MRI, and PET, it is often challenging to differentiate gliomas from other tumors, such as CNS lymphomas and metastatic brain tumors, and even from non-neoplastic diseases such as MS and radiation necrosis. Resection surgery and/or biopsy followed by pathological diagnosis are necessary in such cases. Thus, it is necessary to develop a reliable and less invasive diagnostic tool for gliomas.

One of the main reasons gliomas are difficult to manage is their remarkable invasiveness in the brain parenchyma. Gliomas spread and grow along fibers in the white matter, occasionally invading the other hemisphere via the corpus callosum, and can spread over the entire CNS, including the spinal cord. Therefore, surgical intervention remains limited because tumor tissue cannot be excised totally without causing functional damage to the affected brain structure. Moreover, many therapeutic drugs do not pass through the blood–brain barrier efficiently, and, for more than a decade, none of the targeted drug therapies have been successful in sufficiently prolonging the overall survival of newly diagnosed gliomas, despite the numerous clinical trials worldwide.

In 2021, the WHO Classification of Tumors of the Central Nervous System was revised in its fifth edition (WHO CNS5), which emphasizes diagnosis of gliomas by genes and biomarkers rather than morphological classification by histopathology alone. Gliomas are classified into adult-type and pediatric-type, which differ in susceptibility, prognosis, and biology [34]. Diffuse gliomas are adult-type; these are the most common gliomas. A simplified classification of diffuse gliomas is shown in Figure 3. They are divided into those with isocitrate dehydrogenase (IDH) mutation and those without. IDH is a key enzyme in the TCA cycle. The mutant IDH has a gain-of-oncogenic function, converting α-ketoglutaric acid (α-KG) to 2-hydroxyglutaric acid (2-HG) [35], which is involved in tumorigenesis as an oncometabolite. The IDH mutation is considered the driver and primary mutation in this group of diffuse gliomas, followed by other various mutations and chromosomal abnormalities [36,37]. Tumors with chromosomal abnormalities, such as 1p/19q codeletion, in addition to the IDH mutation, are classified as oligodendroglioma, IDH-mutant, and 1p/19q-codeleted. Furthermore, tumors with mutations in P53 (TP53), a major tumor suppressor gene [38], and α-thalassemia/mental-retardation-syndrome-X-linked (ATRX), which is involved in chromatin remodeling pathways and regulation of telomeres [39,40,41], are classified as astrocytoma, IDH-mutant. These two types of diffuse gliomas are usually less aggressive and slow-growing, and are therefore called low-grade gliomas (WHO grade 2). However, they are to progress into more malignant forms (grades 3 and 4) over time because of further genetic changes. Astrocytoma, IDH-mutant, WHO grade 4 is the final form and the most malignant astrocytoma. Because its histological characteristic is quite similar to glioblastomas, it is called “secondary glioblastoma” as opposed to “primary glioblastoma”. Secondary glioblastomas are malignant, but with a better prognosis compared with primary glioblastoma. Oligodendrogliomas have the best prognosis of all diffuse gliomas, with a life expectancy of over 10 years, as they often respond to chemotherapy and radiotherapy.

Glioblastomas without IDH mutation (glioblastoma, IDH-wildtype, WHO grade 4) are the most malignant and most common form of diffuse glioma. They do not seem to have preceding forms and are characterized by EGFR mutations and/or amplification as a driver mutation. As mentioned above, these gliomas are fatal, despite modern multimodal therapeutics. New diagnostic and therapeutic tools are required to improve outcome and prolong survival.

### 3.2. Relationship between Glioma Growth and PTPRZ

Research on gliomas has focused on protein tyrosine kinase (PTK) signaling pathways, particularly as PTK overactivity has been shown to induce cell growth; indeed, several PTKs have been identified as proto-oncogenes [42]. In recent years, PTPs have also attracted attention [43]. Some PTPs have tumor-suppressing functions, while other PTPs seem to have oncogenic effects [43]; intriguingly, it seems that PPRG is among the former [44], while PTPRZ is among the latter [4,45,46]. PTPRZ mRNA is highly expressed in glioma cells [4,47], and RNA interference or application of a small-molecule inhibitor against PTPRZ reduces tumor growth and cell migration [45,48]. In Table 1, the major proposed roles of PTPRZ in glioma biology are summarized. Recent reports suggest that PTN-PTPRZ signaling stimulates glioma stem cells (GSCs), a subset of neoplastic cells with stem cell-like properties. These GSCs display potent tumor-initiating and maintaining capacities, and are resistant to chemotherapy and radiotherapy [46,49,50]. In a key study, Bao et al. showed that the PTPRZ1-MET (ZM) oncogenic fusion protein is associated with glioma progression [51]. This fusion is caused by a translocation event involving intron 3 or 8 of PTPRZ and intron 1 of MET, is found in about 14% of secondary glioblastomas (i.e., astrocytoma, IDH mutant, WHO grade 4), and causes the activation of MET signaling [52]. In vivo experiments have shown that glioblastomas with ZM fusions have higher expression of genes related to cell proliferation and lower expression of tumor suppressor genes, resulting in lower survival rates [51].

### 3.3. PTPRZ as a Tumor Marker for Glioma

Given the current limitations of available diagnostic tools in clinical practice and the necessity of developing liquid tumor markers specific to gliomas, PTPRZ holds promise as a glial cell tumor biomarker. PTPRZ is cleaved by proteinases extracellularly, and this soluble PTPRZ (sPTPRZ) is released for shedding [12,21]. Recently, we found that both sPTPRZ-long and sPTPRZ-short, derived from PTPRZ-long and PTPRZ-short, respectively, are detectable in the CSF. The CSF concentration of sPTPRZ-long in glioma patients is 10-fold higher than in patients with MS. Therefore, CSF sPTPRZ is a promising diagnostic biomarker for gliomas, and may reduce the need for surgical biopsy and facilitate early management [47]. Furthermore, in situations where CSF samples cannot be obtained preoperatively, the biomarker test can be performed intraoperatively to help the surgeon in deciding whether the lesion should be excised (i.e., high levels indicating glial cell tumors) or not (i.e., low levels excluding glial cell tumors [e.g., lymphoma]). In such a situation, after opening the dura, the surgeon can collect a small amount of CSF and run the test inside the operation room.

Moreover, the new biomarker can help follow the patient’s response to treatment after surgery (e.g., chemotherapy and radiotherapy). In addition, biomarker testing can help detect possible tumor recurrence when levels start to rise again.

One must be cautious that, compared with PTPRZ in glioma cell lines, sPTPRZ in the CSF of glioma patients is heavily glycosylated, and most of the commercially available anti-PTPRZ antibodies are unable to detect CSF sPTPRZ. We found that one of the reactive antibodies detects HNK-1-capped O-Man glycan on PTPRZ [56]. Brain- and glioma-specific PTPRZ glycosylation patterns may therefore help in preparing high-affinity anti-PTPRZ antibodies that selectively detect CSF variants (sPTPRZs).

### 3.4. PTPRZ as a Therapeutic Target for Glioma

Although there are a limited number of basic studies on PTPRZ as a therapeutic targets in vitro and in animals such as mice, there are currently few clinical studies in humans. How PTPRZ overexpression promotes glioma growth remains unclear. Nevertheless, given that PTPRZ-deficient mice appear to develop normally [20], the PTPRZ protein could be a promising therapeutic drug target. Indeed, knockdown of PTPRZ by RNA interference suppresses glioblastoma growth and migration in vitro and in vivo [4,48,49]. A monoclonal antibody against PTPRZ delays glioblastoma growth [55]. Small molecules that inhibit PTPRZ have also been studied, including anti-PTPRZ immunotoxin (7E4B11-SAP), SCB4380, and NAZ2393; these have been shown to suppress glioma tumor cell migration and proliferation, and to inhibit tumor growth in a mouse model of transplanted glioma cells [45,49,57]. PTN and MK, which are members of the pleiotrophin family and PTPRZ ligands, inactivate PTPRZ, thereby inhibiting cell migration and tumorigenicity [49,57]. Even though PTN and MK cannot cross the blood–brain barrier, there is a report of a synthetic molecule that mimics them and is selective for PTPRZ and can penetrate the blood–brain barrier [58]. Some of these substances cannot cross the blood–brain barrier and are inappropriate as therapeutic agents for CNS diseases, and it is therefore necessary to synthesize or discover substances with similar actions. Furthermore, because ZM fusion genes are frequently found in secondary glioblastomas with poor prognosis and higher MET activity, clinical trials of MET kinase inhibitors have been recently initiated [52]. A Phase I study was conducted, and safety and efficacy were confirmed [52]. Future clinical trials are anticipated.

## 4. PTPRZ in Demyelinating Disease

In the vertebrate nervous system, nerve axons are covered by myelin sheaths, which electrically insulate the axons and allow for saltatory conduction, thereby increasing the velocity of signal transmission. In the CNS, myelin sheaths are formed by oligodendrocytes. In demyelinating diseases such as MS, autoimmune responses to myelin component proteins in the CNS are thought to cause inflammatory demyelination, axonal damage, and gliosis. The etiology of MS is not fully understood; however, it is a multifactorial disease, with various risk factors, including female sex, Caucasian race, genetic factors, such as HLA DR15/DQ6 and IL2RA and IL7RA alleles, as well as environmental factors such as Epstein–Barr virus infection, low vitamin D level, and lack of exposure to sunlight [59,60]. Although autoimmune responses specific to MS are not clear, once the blood–brain barrier allows lymphocytes recognizing myelin antigens to enter the CNS, these cells are thought to target myelin sheaths and oligodendrocytes, triggering a series of events that lead to the formation of acute inflammatory demyelinating lesions [59,61]. Interferon-γ (IFNγ)-producing helper T cells (Th1) and interleukin-17 (IL-17)-producing helper T cells (Th17), which specifically respond to myelin sheath antigens, invade the CNS, where they are activated by antigens to produce inflammatory cytokines and chemokines. This results in the mobilization of inflammatory cells such as microglia and macrophages to form inflammatory demyelinating foci [62]. In addition, the involvement of other immune system cells, such as B cells, has been reported [59,63]. In the chronic progressive phase, the innate immune system, including microglia, is activated, chronic inflammation is triggered, and axons are disrupted by the loss of myelin and glial support [64,65]. It is also thought that nitric oxide and reactive oxygen species secreted from microglia damage axonal mitochondria, resulting in demyelination and neurodegeneration [66,67]. The clinical manifestations of MS include visual acuity and visual field disturbances, eye movement disorders, dizziness, dysarthria, muscle weakness, sensory disturbances, cerebellar ataxia, and vesicorectal disturbances. According to the McDonald criteria [61,68], MS is classified into two groups based on clinical course—relapsing–remitting MS, in which patients experience repeated relapses and remissions, and primary progressive MS, in which patients follow a chronic progressive course from the beginning of the disease [69]. About half of relapsing–remitting MS patients progress to secondary progressive MS in 15–20 years [70,71]. The repair of demyelinated MS lesions is carried out by OPCs in a process called remyelination [72]. For remyelination, OPCs must migrate to the demyelinated lesion, and then proliferate and differentiate into oligodendrocytes [73]. Several studies have shown that OPCs are often unable to differentiate in chronic MS lesions [74,75,76]. Table 1 shows the role of PTPRZ in remyelination in MS. It has been reported that oligodendroglial and astrocyte PTPRZ controls OPC differentiation in different ways [53,55]. Oligodendroglial Fyn kinase phosphorylates Rho family GTPases to enhance OPC differentiation [54], while the dephosphorylation of Rho GTPases by PTPRZ results in attenuated OPC differentiation [53]. Accumulating evidence suggests that acute inflammation is required for remyelination, while a chronic inflammatory environment has a deleterious impact on the remyelination process. PTPRZ modified with branched O-Man glycans by the branching glycosylation enzyme GnT-IX is highly expressed by reactive astrocytes in demyelinated lesions [14,15]. Furthermore, a lack of GnT-IX results in reduced astrogliosis and enhanced remyelination in mouse models of demyelination [15], suggesting that GnT-IX-mediated O-Man branching of PTPRZ enhances demyelination by astrocytes. Moreover, astrocytes create a hyaluronan and chondroitin sulfate-rich extracellular environment that prevents remyelination [77,78]. However, it is currently unclear whether PTPRZ is the critical core protein involved in the control of demyelination/remyelination in MS.

## 5. Concluding Remarks

PTPRZ is highly expressed and modified with specific glycans in the CNS. Given that PTPRZ plays a key pathogenetic role in CNS diseases such as gliomas and demyelinating diseases, therapeutics targeting PTPRZ are increasingly sought after. In addition, PTPRZ has attracted attention as a diagnostic marker because CSF sPTPRZ is markedly higher in glioma patients. A liquid biomarker for glioma patients would contribute to the early diagnosis and appropriate treatment of gliomas. Furthermore, PTPRZ negatively affects remyelination in MS, and the genetic deletion of PTPRZ or its glycosylating enzyme enhances remyelination in model mice. The presence of massive glycans on neural PTPRZ hinders the development of diagnostic antibodies; however, brain-specific glycosylation appears to regulate PTPRZ activity, raising the possibility of glycan-targeted therapeutics. PTPRZ is a unique and intriguing molecule closely related to CNS diseases. We anticipate that further research on PTPRZ will ultimately improve the prognosis of CNS diseases.

## Figures and Tables

**Figure 1 ijms-23-04414-f001:**
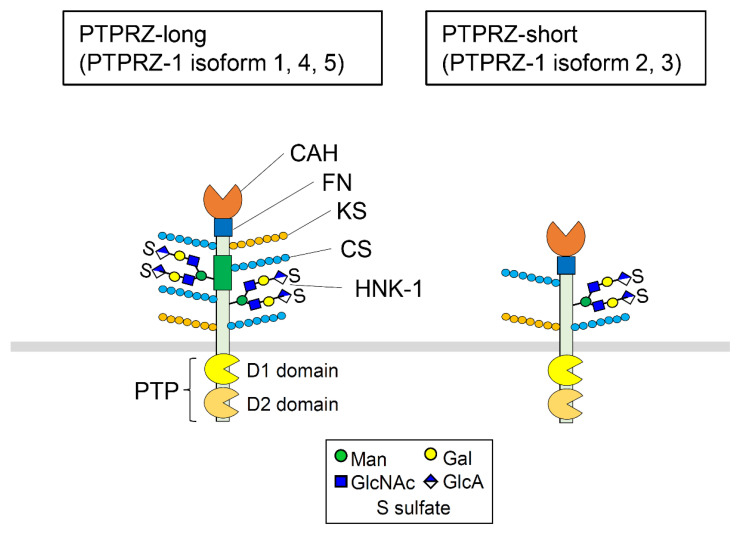
Molecular structure of PTPRZ. PTPRZ has an extracellular carbonic anhydrase-like domain (CAH), a fibronectin-like domain (FN), and two intracellular tyrosine phosphatase domains (PTP), namely D1 and D2. Furthermore, PTPRZ expressed in the central nervous system is highly modified by chondroitin sulfate (CS), keratan sulfate (KS), and HNK-1.

**Figure 2 ijms-23-04414-f002:**
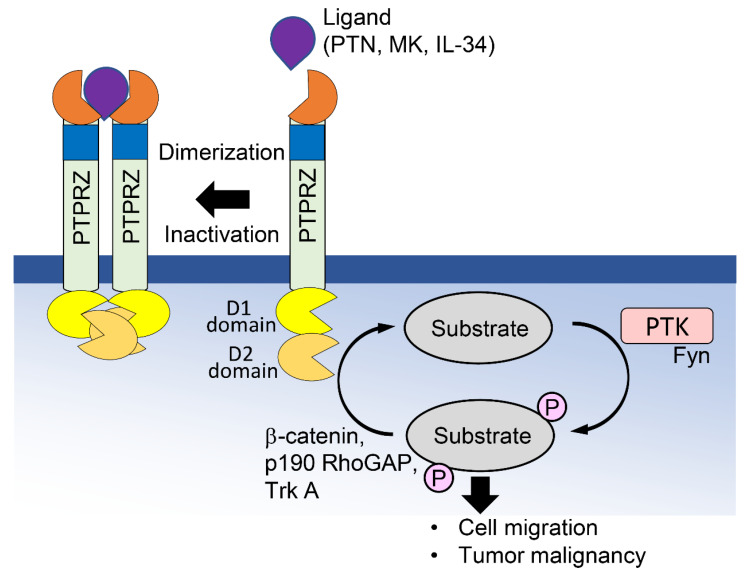
Mechanisms regulating the PTPRZ signaling pathway. PTKs, such as Fyn, phosphorylate downstream signaling molecules, such as β-catenin and p190 RhoGAP, leading to increased cell growth and migration. PTPRZ dephosphorylates these molecules. The binding of ligands induces head-to-toe dimerization of PTPRZ, which results in the masking of the catalytically active D1 domain, leading to its inactivation.

**Figure 3 ijms-23-04414-f003:**
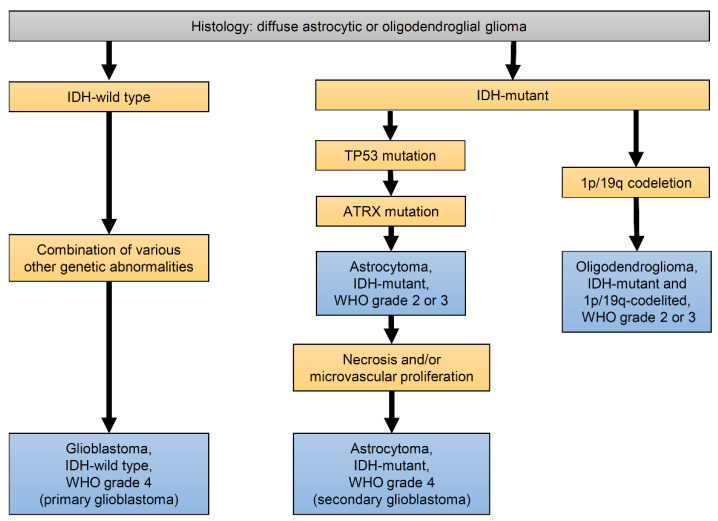
Diagnostic algorithm for major diffuse gliomas in adults based on the 2021 WHO Classification of Tumors of the Central Nervous System. Adult-type diffuse gliomas are thought to be caused by various genetic alterations, such as IDH mutation, 1p/19q codeletion, TP53 mutation, and ATRX mutation. IDH mutation and 1p/19q codeletion are essential in oligodendroglioma. In addition to IDH mutation, TP53 mutation and ATRX mutation occur in astrocytoma, and necrosis is associated with grade 4 astrocytoma. This is referred to as secondary glioblastoma in the 2016 WHO classification. Glioblastoma is caused by a combination of various other genetic abnormalities without IDH mutation.

**Table 1 ijms-23-04414-t001:** Role of PTPRZ in glioma and multiple sclerosis.

**Glioma**
PTPRZ has an oncogenic effect [4,45,46]. Activation of PTPRZ promotes tumor growth and tumor cell migration [45,48]. PTPRZ stimulates glioma stem cells (GSCs) to maintain tumorigenicity [46,49,50]. PTPRZ1-MET(ZM) oncogenic fusion protein is associated with glioma progression [51].
**Multiple Sclerosis**
PTPRZ dephosphorylates the Rho GTPase in oligodendroglia, inhibits oligodendrocyte precursor cell (OPC) differentiation, and prevents remyelination [53,54,55]. PTPRZ modified with branched O-Man glycosyl groups by GnT-IX promotes demyelination by astrocytes [14,15].

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
