# Peer review of "Protein Tyrosine Phosphatase Receptor Type Z in Central Nervous System Disease"

_ijms, 2022, doi:10.3390/ijms23084414_

Round 1
Reviewer 1 Report
The manuscript describes a very interesting protein which can be potential biomarker in diagnostic and molecular target for therapy of gliomas and MS. Thus from clinical point of view, the subject of the publication is very attractive. The manuscript is very consistent and concise, which is an advantage, but the chapter on glioblastoma therapy should be extended. In this chapter 13 publications were cited and only 3 sentences were written. This part of the manuscript is very important and requires more details. This will improve the publication and it will be more valuable for readers. Besides, the title should be changed as MS affects not only brain but also spinal cord, so the title should as follows: Protein tyrosine phosphatase receptor type Z in CNS disease
Author Response
Point-by-point responses to Reviewer #1’s comments
Thank you very much for your kind suggestions and comments.
Comment: The manuscript describes a very interesting protein which can be potential biomarker in diagnostic and molecular target for therapy of gliomas and MS. Thus from clinical point of view, the subject of the publication is very attractive. The manuscript is very consistent and concise, which is an advantage, but the chapter on glioblastoma therapy should be extended. In this chapter 13 publications were cited and only 3 sentences were written. This part of the manuscript is very important and requires more details. This will improve the publication and it will be more valuable for readers. Besides, the title should be changed as MS affects not only brain but also spinal cord, so the title should as follows: Protein tyrosine phosphatase receptor type Z in CNS disease
Response : We agree with the title change as the reviewer proposed. In the revised manuscript, we have inserted several underlined sentences in the chapter “PTPRZ as a therapeutic target for glioma” as follows:
Page 6-7, line 207 – 227.
3.4. PTPRZ as a therapeutic target for glioma
Although there are a limited number of basic studies on PTPRZ as a therapeutic target in vitro and in animals such as mice, there are currently few clinical studies in humans. How PTPRZ overexpression promotes glioma growth remains unclear. Nevertheless, given that PTPRZ-deficient mice appear to develop normally [20], the PTPRZ protein could be a promising therapeutic drug target. Indeed, knockdown of PTPRZ by RNA interference suppresses glioblastoma growth and migration in vitro and in vivo [4, 48, 49]. A monoclonal antibody against PTPRZ delays glioblastoma growth[55]. Small molecules that inhibit PTPRZ have also been studied, including anti-PTPRZ immunotoxin (7E4B11-SAP), SCB4380 and NAZ2393, which have been shown to suppress glioma tumor cell migration and proliferation, and to inhibit tumor growth in a mouse model of transplanted glioma cells[45, 49, 57]. PTN and MK, which are members of the pleiotrophin family and PTPRZ ligands, inactivate PTPRZ, thereby inhibiting cell migration and tumorigenicity [49, 57]. Even though PTN and MK cannot cross the blood–brain barrier, there is a report of a synthetic molecule that mimics them and is selective for PTPRZ and can penetrate the blood–brain barrier[58]. Some of these substances cannot cross the blood–brain barrier and are inappropriate as therapeutic agents for CNS diseases, and it is therefore necessary to synthesize or discover substances with similar actions. Furthermore, because ZM fusion genes are frequently found in secondary glioblastomas with poor prognosis and higher MET activity, clinical trials of MET kinase inhibitors have been recently initiated [52]. A Phase I study was conducted, and safety and efficacy were confirmed [52]. Future clinical trials are anticipated.
Reviewer 2 Report
In this review article, the authors summarized recent reports and findings about the essential roles of protein tyrosine phosphatase receptor type Z (PTPRZ) in regulation of biological features of glioma, one of the most common intracranial malignancies, and multiple sclerosis (MS), a representative intracranial demyelinating disease.
# Comments:
- From the volume and intensity of the contents, I consider it is appropriate to treat this article as “Mini review” but not “Review”.
- More general information about the molecular signaling regulated by PTPRZ should be summarized and illustrated (in chapter 2).
- The authors well described about general knowledge of gliomas (chapter 3.1); however, in contrast, description about general knowledge of MS is considered insufficient (chapter 4). Therefore, the authors should describe about more detailed backgrounds of MS in chapter 4, especially about molecular pathogenesis.
- To make this article more comprehensive, the authors should summarize the roles of PTPZR in glioma (chapter 3.2 and 3.3) and MS (chapter 4) into one table, respectively, with the number of references.
- In line 167 – 169, I consider it is not adequate to discuss the role of glycosylation of PTPRZ by exemplifying extracellular proteoglycans; many plasma membrane localizing proteins are known as undergone glycosylation and exerts various biological function such as intracellular shuttling. Therefore, the authors should discuss this part more carefully by showing appropriate references.
Round 2
Reviewer 2 Report
The authors responded to all my questions and requests appropriately.